# *Arabidopsis* Flowers Unlocked the Mechanism of Jasmonate Signaling

**DOI:** 10.3390/plants8080285

**Published:** 2019-08-14

**Authors:** John Browse, James G. Wallis

**Affiliations:** Institute of Biological Chemistry, Washington State University, Pullman, WA 99164, USA

**Keywords:** *Arabidopsis*, flower development, jasmonate, JAZ, transcript profiling

## Abstract

The *Arabidopsis* male-sterile phenotype has been a wonderful model for jasmonate action in plants. It has allowed us to identify transcription factors that control gene expression during stamen and pollen maturation and provided for the discovery of the JAZ repressor proteins and the mechanism of jasmonate signaling. More recently, it has revealed intriguing details of the spatial localization of jasmonate synthesis and perception in stamen tissues. The extensive and thoughtful application of protein–protein interaction assays to identify JAZ-interacting partners has led to a much richer appreciation of the mechanisms by which jasmonate integrates with the actions of other hormones to regulate plant growth and physiological responses. This integration is strikingly evident in stamen and pollen development in *Arabidopsis*, which requires the actions of many hormones. Just as importantly, it is now evident that jasmonate has very different actions during flower development and reproduction in other plant species. This integration and diversity of action indicates that many exciting discoveries remain to be made in this area of jasmonate hormone signaling and response.

## 1. Introduction

Jasmonate (JA) hormone controls many aspects of plant biology, ranging from stress responses to development. These include defense against herbivores and microbial pathogens, as well as responses to ultraviolet radiation, drought, ozone, and other abiotic stresses [1,2,3,4,5]. In healthy plants, JA regulates aspects of carbon partitioning, reproductive development, and senescence [6,7,8,9]. Jasmonate signaling leads to large-scale changes in gene expression, and hundreds of downstream genes have been identified [10]. Mutants of *Arabidopsis* have provided the means to dissect the biochemistry and cell biology of JA synthesis [2,11], and the conversion of JA to the active hormone jasmonoyl-isoleucine (JA-Ile) [12,13]. In addition, *Arabidopsis* mutants were key tools in the discovery of the central role of JA in marshalling plant defenses against herbivores and necrotrophic pathogens [14,15]. Details of many of these topics are covered in other articles in this issue. In this review, we will describe the discovery of the fact that JA is required for stamen and pollen maturation in *Arabidopsis*, and how investigations of the resulting male-sterile phenotype contributed to our current mechanistic understanding of JA signaling.

## 2. Two Arabidopsis Mutants Reveal Jasmonate’s Roles in Stamen and Pollen Maturation

Coronatine is a phytotoxin virulence factor that is produced by several bacteria including *Pseudomonas syringae* pv *tomato* DC3000 (*Pst* DC3000) [16]. It has been known for some time that coronatine acts through activation of JA signaling [7,16]. In 1994, a screen for mutants of *Arabidopsis* resistant to coronatine identified the *coi1* (*coronatine insensitive1*) mutant that was also resistant to JA and male sterile (Figure 1) [7]. At about the same time, our laboratory was studying photosynthesis in mutants of *Arabidopsis* with reduced levels of polyunsaturated fatty acids in chloroplast membrane lipids. The very high proportions of trienoic fatty acids (including linolenate) found in chloroplast membranes of all higher plants suggest that these lipid structures might be essential for photosynthesis. To test this notion, we produced a *fad3–fad7–fad8* triple mutant (*fad* = *fatty acid desaturation*) that completely lacked trienoic fatty acids. To our surprise, photosynthesis at 22 °C in the triple mutant was barely affected and vegetative growth of mutant plants was identical to wild-type controls; however, the mutant plants were male sterile [17]. Importantly, treatment of *fad3–fad7–fad8* flowers with JA, which is derived from linolenate, rendered them fertile (Figure 1) [17].

Characterization of flowers in the *Arabidopsis fad3–fad7–fad8* triple mutant identified three features of the male-sterile phenotype: (1) floral organs develop normally within the closed bud, but the anther filaments do not elongate to position the locules above the stigma at the time of flower opening; (2) the anther locules do not dehisce at the time of flower opening (although limited dehiscence occurs later); and (3) even though pollen on mutant plants develops to the trinucleate stage, the pollen grains are predominantly (>97%) inviable. Treatment of flower buds with JA corrected all three defects, providing for rates of pollen germination equivalent to wild type in in vitro tests, and abundant seed set on treated plants [17]. Restoration of fertility was stage specific—only flower buds corresponding to the transition between stages 11 and 12 in floral development, immediately before flower opening, responded to JA [17].

## 3. A Transcriptional Cascade Directing Pollen and Stamen Development

The ability to screen male-sterile mutants for complementation by JA treatment led to the isolation of a mutant in *opr3*, the gene that encodes the isomer-specific 12-oxo-phytodienoic acid reductase that acts in the JA synthesis pathway [18,19]. Although *opr3* plants do not accumulate JA, they express some defense genes and are resistant to insect attack [20]. Other mutants in JA synthesis are also male-sterile [21]. Since the *opr3* mutant could be made fertile by JA application, we carried out transcriptional profiling and followed the time course of gene-expression changes in *opr3* stamens for 22 h following JA treatment [22]. Analysis of data from four time points following JA or control OPDA treatment (0.5, 2, 8, and 22 h) identified 1296 genes that had expressions that were specifically altered by JA treatment. These genes amount to a stamen-specific JA transcriptome that is quite different from the transcriptome involved in defense responses. Most (70%) of the genes were expressed in the parental tissues rather than the pollen. Using bioinformatics, we mapped many of the upregulated genes to metabolic pathways that are likely to be induced during stamen maturation. Pathways identified include the synthesis of terpenoid volatiles, which have roles in attracting insects, and synthesis of wax compounds that are functionally important components of the pollen kit [22]. Other aspects of flower development that are affected in JA mutants include petal elongation, stigma papillae, and carpel vasculature [1,7,22].

We identified 13 transcription factors induced by JA that potentially act in the control of stamen maturation. Two of these transcription factors, MYB21 and MYB24, are the only members of subgroup 19 of the R2R3 MYB family of proteins. The *MYB21* gene had previously been identified in Giltsu Choi’s laboratory as being ectopically expressed in vegetative tissues of the *constitutive photomorphogenic1* (*cop1*) mutant [23], and we collaborated to investigate *myb21* and *myb24* mutants. The null *myb21* mutant exhibited a JA-insensitive flower phenotype, and male fertility was significantly reduced. The phenotype of a *myb24* mutant was akin to wild type, but examination of double-mutant *myb21–myb24* plants established that inclusion of the *myb24* mutation in the plants aggravated all three aspects of the *myb21* phenotype [22]. These results show that JA induces MYB21 and MYB24 activity and thereby controls essential aspects of stamen development. Since this discovery, additional studies have provided more details about the induction and actions of other transcription factors in addition to MYB21 and MYB24 that coordinate anther and pollen maturation and initiate anther filament elongation, ensuring that the mature anther is correctly placed relative to the stigma when the anthers dehisce [24,25].

## 4. Discovery of the JAZ Repressors

In 1998, when map-based cloning identified the *COI1* locus, it was revealed to encode an F-box protein [26] expected to participate in a Skp/Cullin/F-box complex (SCF^COI1^) acting as an E_3_ ubiquitin ligase [27]. In company with the phenotype of *coi1* mutants, this discovery suggested that SCF^COI1^ ubiquitination of proteins was essential to JA hormonal activity and required for all JA responses. With the demonstration that COI1 was an F-box protein, two questions became central to the analysis of JA signaling for the next decade: (1) What proteins are targets of the SCF^COI1^ complex? (2) How does (presumed) ubiquitination and degradation of these proteins via 26S protease action enable JA hormone activity? Initially, the searches for SCF^COI1^ targets using genetic approaches were unavailing. Wide-ranging genetic screens failed to detect candidates for SCF^COI1^ targets; searches for positive effectors [28,29,30], negative effectors [28,31], and components downstream of COI1 [32], as well as searches for COI1-interacting proteins [33,34], were all unsuccessful. Investigations identifying genes whose expression responded to JA [35] or that relied on COI1 [36] were likewise unsuccessful in detecting candidates, and failed to provide indications about the exact nature of the interaction between SCF^COI1^ and transcription of genes responsive to JA regulation.

The search finally ended when transcripts of *opr3* stamens were profiled, resulting in identifying the JAZ protein family and discovery of their role in JA signaling. Immediately following JA application (at the 0.5 h time point), only 31 genes showed significant induction. Amongst these, eight encoded proteins with no known function, and the predicted protein sequences of the eight were used to query the Conserved-Domain Database at NCBI using Reverse Position-Specific BLAST (RPS-BLAST) [37]. Each of the eight contained a 28 amino-acid domain with no known function, but occurring in ZIM, a protein considered likely be a transcription factor [38], and so were named JAsmonate Zim-domain proteins. The eight *JAZ* genes first identified were induced in stamens 6 to 40 fold after JA treatment, and were also found to be expressed widely during plant development. Experimental data using 7 day-old seedlings treated with 10 μM JA indicated strong, 8 to 60 fold induction of all eight genes in the first half hour after treatment [39].

There are now 13 recognized JAZ proteins in *Arabidopsis* [39,40,41,42]. Homology among them is principally confined to two domains. The ZIM domain of 27 amino acids is located near the middle of the peptide sequence; it contains a TIFY motif (TIF(F/Y)XG) [43] and two invariant alanine residues. The Jas domain of 22 amino acids lies close to the C-terminus and is strongly conserved across the JAZ family. In some JAZ, a less conserved N-terminal region contains an EAR (ethylene response factor associated amphiphilic repression) motif and/or binding determinants for some transcription factors [44,45].

Jasmonoyl-isoleucine and coronatine promote binding of the JAZ proteins to COI1 through the Jas domain [39,46,47,48]. Deletion of the Jas domain or specific mutations within it prevent binding, so that JAZΔJas proteins become constitutive repressors [39,40,46,47,49]. Expression of these altered JAZ proteins in plants induced a dominant JA-resistant phenotype [39,40,44,45]. Experiments with plants expressing JAZ1-GUS [39] and JAZ3-GFP proteins [40] demonstrated that turnover of these proteins does indeed require COI1 and the 26S proteasome, as well as the Jas domain of the JAZ proteins. The rapid induction of some of the *JAZ* genes in response to JA signaling likely results in extensive re-synthesis of the JAZ repressors and attenuation of the JA response soon after passage of the signal [44,45]. However, genes that encode enzymes of JA-Ile synthesis and the gene encoding MYC2 are also strongly induced on a similar timescale. Transcriptional induction for all these genes occurs in the presence of cycloheximide indicating that it is not dependent on the synthesis of intervening transcription factors [50]. Thus, genes involved in generation and transmission of the JA signal are induced alongside genes encoding the repressors.

These findings suggested a model in which JAZ proteins are repressors that bind to transcription factors and prevent the transcription of JA-responsive genes by recruiting co-repressor proteins into an inhibited complex. It is now known that the ZIM domain recruits corepressors of the TOPLESS family via the EAR-domain protein NINJA [51] and also, in some cases, directly via an EAR domain on the JAZ protein. Three JAZ proteins have been shown to interact with the histone deacetylase enzyme, HDA6 [52], suggesting that JAZ repression may also involve chromatin remodeling.

In the ten years since these foundational discoveries, genetic and molecular approaches have elaborated many details of JA signaling and response that modify a wide range of growth and defense processes in all tissues of *Arabidopsis* and other plants [44,45,53,54]. Perhaps more importantly, screening and identification of JAZ-interacting proteins have led to discoveries that demonstrate how completely JA hormone action is integrated with the actions of other hormones, including gibberellin, abscisic acid, salicylic acid, ethylene, and auxin [55,56]. This is true in flower development as much as in any other aspect of plant development and physiology [57,58].

## 5. Jasmonate Receptor Crystal Structure

The identification of the JAZ proteins and evidence that the Jas domain is the site of binding to COI1 provided the tools necessary to solve the structure of JA-Ile bound to its receptor [46]. Crystal structure analysis revealed that JA-Ile (or the analogue, coronatine) is oriented vertically in its interaction with 11 COI1 residues of the ligand-binding pocket of this F-box protein class, which is formed by a leucine-rich-repeat/loop structure. Three other residues form a cavity that binds only the (3R,7S) active isomer of JA-Ile [59], but does not accept the (3R,7R) inactive isomer. While the JA keto group and Ile carboxyl group are left exposed and accessible to interaction with JAZ protein through its Jas domain, the rest of the JA-Ile ligand is concealed from interaction by the COI1 residues surrounding the binding site. Surprisingly, experiments assessing the strength of the interaction using radioligand biding demonstrated very weak binding between JA-Ile and COI1, although the crystal structure indicated many interactions were probable. However, when JAZ protein was incorporated into such an assay, specific binding increased 50 fold, demonstrating that both the JAZ and COI1 proteins function as interdependent co-receptors in perception of JA hormone [46]. The structural elements of the Jas domain which bind COI1 and JA-Ile (residues 200–220 in the JAZ1 protein) are in two separate regions of the domain. The N-terminal sequence, ELPIA, is a loop-forming structure interacting with both the carbonyl group found on JA-Ile as well as identified COI1 residues, with the result that the binding site of the JA-Ile co-receptor is closed to hormone access. The C-terminal portion of the Jas peptide sequence produces an α-helical structure interacting with amino acids lining the central channel of the leucine-rich repeat ring formed by COI1, and this supplementary binding is indispensable for the co-receptors to function.

Because the related TIR1 receptor was found to contain a molecule of inositol hexakisphosphate (IP_6_) [60], it was not particularly surprising to see evidence for a similar molecule in crystals of recombinant COI1. However, the molecule in COI1 was found to be inositol 1,2,4,5,6-pentakisphosphate (IP_5_) and to be located close to the JA-Ile binding pocket where one of the phosphate residues coordinates three arginine residues of COI1 and Arg206 of the Jas peptide, which are all also involved in interactions with JA-Ile. Consistent with this structure, removal of IP_5_ by dialysis inactivated the receptor complex and the inactive form could be reactivated by addition of IP_5_ [46]. Subsequent studies have concluded that, in plants, an inositol pyrophosphate (IP_8_) is the physiological cofactor in COI1 that has a key role in assembly and functioning of the hormone receptor [61,62].

## 6. What Are the Sites of JA Perception in the Stamen?

The male-sterile phenotype of *Arabidopsis* mutants defective in JA synthesis or perception also allowed us to answer the question of where in the stamen JA signaling is required to correct the defects in filament elongation, anther dehiscence, and pollen function. To do this, we constructed a DNA sequence encoding a COI1-YFP fusion protein and expressed it in *coi1* mutant plants using different tissue-specific promoters [63]. The yellow fluorescent protein (YFP) allowed localization of the COI1 protein by confocal microscopy. As a control, we first used the endogenous *COI1* promoter. Confocal microscopy confirmed broad expression of the COI1-YFP protein throughout the stamen and anther tissues of the transgenic plants, and fertility was fully restored, indicating the COI1-YFP fusion protein was functional.

Given that stamen filament elongation does not occur in JA mutants, the filament was expected to be a target of JA signaling. During their characterization of the JA biosynthesis mutant *defective anther dehiscence1* (*dad1*), Ishiguro et al. [21] discovered that the filament is likely the major or sole site of JA synthesis and suggested that the filament may be the only essential target of JA for stamen and pollen maturation and function. The model proposed by Ishiguro et al. showed JA signaling in the upper filament promoting sucrose import into the filament. The lower water potential then drives water transfer from the anther locules and endothecium into the filaments. Dehydration of the anther tissues and pollen could then lead to maturation of pollen grains and promote anther dehiscence by desiccation of the endothecium, while the filament undergoes elongation [21]. When we expressed a *DAD1::COI1-YFP* transgene in *coi1* plants, the YFP signal was confined to the filament and filament elongation did occur, although it was delayed relative to wild-type controls. However, anther dehiscence did not occur, and pollen remained inviable [63]. These results indicate that there must be sites of JA perception in the anther and suggest that JA synthesized in the filament may be transported into the anther to initiate JA responses there. Because the tapetum has essential roles in pollen maturation, we next expressed our *COI1-YFP* construct behind the tapetum-specific *A9* promoter. In this case, strong YFP fluorescence was seen in tapetal cells, but no aspect of the *coi1* phenotype was altered and the transgenic plants were completely sterile, indicating that any JA-dependent responses in the tapetum are insufficient to affect any aspect of the male-sterile phenotype.

In contrast to this result, the *WUS (WUSCHEL)* and *LTP1 (LIPID TRANSFER PROTEIN1)* promoters that provided COI1-YFP expression in the stomium and epidermis restored both pollen function and anther dehiscence. In particular, the *LTP1::COI1-YFP* construct that is only expressed in the epidermal cells of the anther and filament resulted in fully fertile plants, although both filament elongation and anther dehiscence were delayed relative to wild-type controls (Figure 2) [63].

These results indicate that JA perception only in the epidermis of the anther and filament is sufficient for anther dehiscence and for pollen viability. The stomium is comprised of epidermal cells, so this readily explains JA control of anther dehiscence [18]. Elongation of the filament may also be rationalized since tissue growth in other plant organs has been shown to be controlled by the epidermis [64]. Epidermal control of pollen viability by JA is perhaps more surprising, but it is consistent with the observation that, genetically, it is the sporophyte tissue that mediates JA-regulated male fertility. When JA signaling is blocked (as in the *coi1* mutant), some sporophytic process compromises pollen viability. These results demonstrate that triggering JA-receptor action solely in the epidermis provides for pollen viability. One possibility is that JA signaling in the epidermis leads to water loss from the pollen (as well as the anther tissues), and this triggers the final maturation of the pollen. Desiccation during later stages of anther development has been observed in previous studies and was proposed to account for JA-dependent pollen maturation [21]. Alternatively, the anther epidermis may produce a transmissible signal in response to JA that is responsible for final pollen maturation. This scenario is similar to the cell non-autonomous complementation of leaf expansion by epidermal brassinosteroid perception [64] and is equally consistent with the results of our COI1-YFP complementation studies [63].

## 7. Jasmonate-Regulated Processes in Flower Development Are Very Different in Other Angiosperms

The reproductive phenotype of *Arabidopsis* JA mutants has been a rewarding source of discoveries in many aspects of JA synthesis and signaling. However, the story of JA’s role in plant reproduction becomes even more interesting as we consider the role of JA in flower development beyond this model species. Many of the actions of JA in herbivore and pathogen defense and in vegetative plant growth are broadly observed across angiosperm species. However, as Carl Linnaeus observed, evolution has worked diligently to tweak the reproductive organs and processes of plants because they are the major basis for speciation. Thus, a tomato mutant (*jai1*) defective in COI1 is not male sterile, although pollen viability is lowered with premature dehydration and dehiscence of the anther [65]. Instead *jai1* plants are defective in maternal control of seed development [66]. The tomato MYB21 homologue has been shown to be a component of JA signaling in tomato reproduction [67], suggesting a core pathway similar to that in *Arabidopsis* is providing for a rather distinct output.

In the monocots, the role of JA in floral development is very different from these two dicot species. In rice, the floral spikelet is a condensed branch in which florets are enclosed in leaf-like glumes that open to allow release of wind-dispersed pollen and pollination of the pistil. In this system, investigations of several mutants, including *extra glume1* (*eg1*) encoding a rice DAD1 homologue and *eg2* encoding an OsJAZ1 with altered Jas domain [68], indicate that JA acts very early in spikelet development. Although the phenotypes of *eg1* and *eg2* are rather mild, possibly due to the redundancy in the encoded gene functions, additional genetic and molecular analyses suggest a model in which JA-mediated degradation of OsJAZ allows the rice MYC2 homologue to induce expression of OsMADS1, OsMADS7, and OsMADS8 transcription factors that initiate and control spikelet development [68,69]. These rice MADS box proteins are homologues of the SEPALLATA family of transcription factors in *Arabidopsis* that are required for expression of B- and C-class homeotic regulators of floral development [70,71]. So, in rice, the core JA-signaling module, JAZ/JA-Ile/COI1/MYC2, appears to have been redirected (relative to *Arabidopsis*) to control fundamental processes at the very start of spikelet and floret development, rather than being required only for the final stages of stamen and pollen maturation.

In maize, separate male and female spikelets of flowers form, both at the topmost male tassel as well as at the lateral meristems responsible for producing the female ears. Despite their different physical appearance, maize flowers form the same elemental pattern as do other flowering plants. Their organs form in four spirals, and both the identity of the organs and their development are controlled by collaboration among homeotic genes in overlapping tripartite regions of the floral meristem denoted A, B, and C [72,73]. When the programmed elaboration of ABC is completed, the male tassels and female ears are barely distinguishable, each having recognizable onset of both pistil and stamen development. Differentiation occurs as the flowers mature, when nascent pistils in the tassel are aborted by a cell death process, while in the ear, stamens undergo arrested cell cycles and deterioration of cells [74,75]. Among the abundant mutants altering sex determination in maize [73,76,77] are those termed *tasselseed* (*ts*), whose typically staminate tassels instead develop pistillate florets. The *ts1* mutations are recessive and have no effect on early spikelet and floret development, but instead result in a completely feminized tassel. When *ts1* was cloned [77], it turned out to encode a 13-lipoxygenase, predicted to localize in the chloroplast, that catalyzes polyunsaturated fatty acid peroxidation. Because the synthesis of JA occurs by means of a 13-lipoxygenated product, 13-hydroperoxylinolenic acid [11], the clear implication was that JA signaling could be important in controlling the pathway identified by the *tasselseed* mutant. Examination of JA levels of tassels from plants homozygous for *ts1-ref* revealed that the concentration was less than 10% of the level obtained from measurements of wild-type tassels. Further, the mutant *tasselseed* phenotype was partially rescued when tassels of the mutant were treated with 1 mM JA [77]. Some other *tasselseed* mutants have also been found defective in JA signaling in the developing tassel [77,78,79,80].

Thus, with four model plant species studied, we have four different stages of flower and reproductive development targeted by JA hormone signaling. What might the rest of the plant kingdom reveal?

## Figures and Tables

**Figure 1 plants-08-00285-f001:**
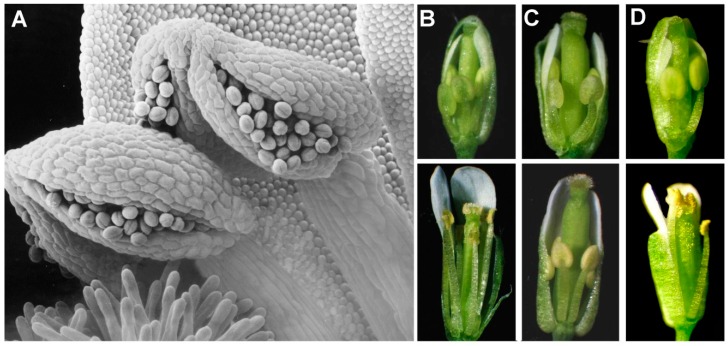
Two *Arabidopsis* mutants that unlocked the mechanism of jasmonate signaling. (**A**) Self-pollinating species like *Arabidopsis* depend on coordinated development of the anther, the pollen, and the anther filament to ensure that the mature anther is positioned above the stigma at the time of anther dehiscence and pollen release. (**B**) In wild-type flowers, filament elongation, pollen maturation, and dehiscence all depend on jasmonate signaling. (**C**) The *coi1* (*coronatine insensitive1*) mutant is male-sterile and does not respond to jasmonate treatment. (**D**) In the *fad3–fad7–fad8* triple mutant (*fad* = *fatty acid desaturation*) (and other jasmonate-synthesis mutants), filament elongation, anther dehiscence, and pollen function are all defective as in *coi1*, but treating flower buds with jasmonate makes the flowers as fertile as wild type. In (**B**–**D**), the upper panels show stage 11 flowers and the lower panels stage 15.

**Figure 2 plants-08-00285-f002:**
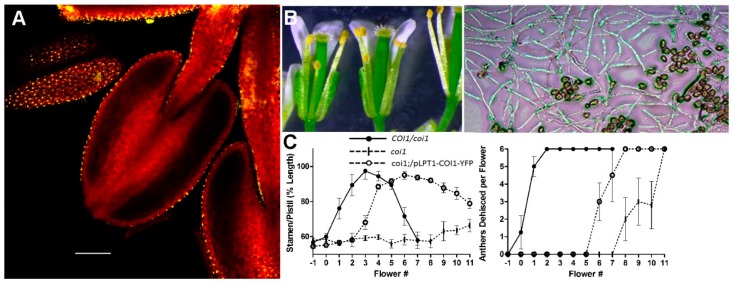
Tissue-specific expression of a *COI1-YFP* transgene rescues *coi1* sterility. (**A**) *coi1* plants expressing a *LTPpro::COI1-YFP* gene show the YFP signal only in epidermal cells of the filament and anther. (**B**) In the transgenic plants, filament elongation, anther dehiscence (left), and pollen-tube germination and growth (right) are all sufficient to restore fertility, although (**C**) filament elongation and anther dehiscence are delayed compared to wild type. (Adapted from Reference [63], with permission.).

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
