# Peer review of "Arabidopsis Flowers Unlocked the Mechanism of Jasmonate Signaling"

_plants, 2019, doi:10.3390/plants8080285_

Round 1

Reviewer 1 Report

Browse and Wallis provided with their review a very nice, comprehensive and at some points very personal summary about function and perception of jasmonic acid (JA). Starting with the model plant Arabidopsis, they give an overview about the role of JA in flower development shown by mutants defective in either JA biosynthesis or perception. Use of both types of mutants led to the discovery of the JA co-receptor COI1 and JAZ, which both were described in detail. Referring to their lab’s own work, the site of JA perception in stamen is summarized here including a very informative figure. The review ends with highlighting the quite different roles of JA in flower development of other species, thereby describing the female phenotype of JA-insensitive tomato, the role of JA in spikelet development of rice and the feminized tassels in JA-deficient maize mutants. With this review, the reader gets an excellent overview about the diverse function that JA has in flower development of different species.

There are only few very minor points/typos, which should be corrected before publishing this review:

Lines 169-170: If space permits, it may be useful to cite the paper by Fonseca et al. (2009, Nature Chem. Biol. 5, 344-350), in which it was shown that (+)-7-iso-jasmonoyl-L-isoleucine is the bioactive isomer.

Lines 259-263: It might be informative to mention that JA is also involved in regulation of the stamen development in tomato (see Dobritzsch et al., 2015, BMC Biol 13, 28). Here, the effects of JA insensitivity are just opposite to that in Arabidopsis, because stamen of the tomato mutant jai1 show premature water loss and dehiscence.

Figure 1B-D: Please, give the stages of floral development for the upper and the lower raw of photographs to make comparison more convenient.

Lines 37 and 46: Since there are two authors of this review, first person singular (I, my lab) should be replaced by first person plural (we, our lab).

Line 115: please, insert a comma after [28-30].

Line 140: please, insert a dot after [39,40,46,47,49].

Line 164: please remove the dot.

Reference list: In the title of some papers (e.g., #9, #10, #24), most words start with upper case letters, which is not necessary. Only the first word of the title should begin with an upper case letter. In addition, species names should be in italics.

Author Response

Lines 169-170: If space permits, it may be useful to cite the paper by Fonseca et al. (2009, Nature Chem. Biol. 5, 344-350), in which it was shown that (+)-7-iso-jasmonoyl-L-isoleucine is the bioactive isomer.

We have added this reference

Lines 259-263: It might be informative to mention that JA is also involved in regulation of the stamen development in tomato (see Dobritzsch et al., 2015, BMC Biol 13, 28). Here, the effects of JA insensitivity are just opposite to that in Arabidopsis, because stamen of the tomato mutant jai1 show premature water loss and dehiscence.

We have added text and this reference

Figure 1B-D: Please, give the stages of floral development for the upper and the lower raw of photographs to make comparison more convenient.

We have added the stages to the figure legend

Lines 37 and 46: Since there are two authors of this review, first person singular (I, my lab) should be replaced by first person plural (we, our lab).

We have corrected this.

Line 115: please, insert a comma after [28-30].

We have corrected this.

Line 140: please, insert a dot after [39,40,46,47,49].

We have corrected this.

Line 164: please remove the dot.

We have corrected this.

Reference list: In the title of some papers (e.g., #9, #10, #24), most words start with upper case letters, which is not necessary. Only the first word of the title should begin with an upper case letter. In addition, species names should be in italics.

We have corrected the errors

Reviewer 2 Report

In this paper, Browse and Wallis summarized a long research history and recent progresses of JA action in flower development. The story began with two Arabidopsis mutants, coi1 and fad3 fad7 fad8, showing characteristic defects in filament elongation, anther dehiscence, and pollen maturation. They mentioned opr3 and other JA biosynthesis mutants showing similar phenotypes and introduced MYB21 and MYB24 transcription factors working downstream of JA signaling. Then they described detailed history from JAZ discovery to its important function in JA signaling, along with the finding of a JA active form, JA-Ile. In the latter part of this paper, thay focused on an interesting discovery of JA perception sites in stamens and discussed distinct JA function in non-Arabidopsis plants, such as tomato, rice, and maize. Although the topics selected in this manuscript were rather dominated by the authors' own studies, it is not a large problem, because they are important and valuable for readers. However, by the subject of this paper, I still think the authors can add a small but important topic that JA effects on petal elongation in brassicaceous and solanaceous plants (1, 2, 3, 4). Please consider. 1. Hatakeyama et al. (2003) Mol. Breed. 11, 325-336 2. Reeves et al. (2012) PLoS Genetics 8, e1002506 3. Stitz et al. (2014) Plant Cell 26, 3964-3983 4. Niwa et al. (2018) Biosci. Biotechnol. Biochem. 82, 292-303. Followings are (typographical) errors. L37 "I" should be "We". L115 "for for" (duplicated) L140, 146 Period missing after "]".

Author Response

We have added text describing the additional effects of JA on petal elongation and other aspects of flower development (l. 90-91), and corrected the other errors.